# Study on the Magnetic Noise Characteristics of Amorphous and Nanocrystalline Inner Magnetic Shield Layers of SERF Co-Magnetometer

**DOI:** 10.3390/ma15228267

**Published:** 2022-11-21

**Authors:** Ye Liu, Hang Gao, Longyan Ma, Jiale Quan, Wenfeng Fan, Xueping Xu, Yang Fu, Lihong Duan, Wei Quan

**Affiliations:** 1School of Instrumentation and Optoelectronic Engineering, Beihang University, Beijing 100191, China; 2Zhejiang Provincial Key Laboratory of Ultra-Weak Magnetic-Field Space and Applied Technology, Hangzhou Innovation Institute, Beihang University, Hangzhou 310051, China; 3Research institute for Frontier Science, Beihang University, Beijing 100191, China

**Keywords:** soft magnetic material, amorphous, nanocrystalline, SERF co-magnetometer, magnetic shield

## Abstract

With the widespread use of magneto-sensitive elements, magnetic shields are an important part of electronic equipment, ultra-sensitive atomic sensors, and in basic physics experiments. Particularly in Spin-exchange relaxation-free (SERF) co-magnetometers, the magnetic shield is an important component for maintaining the SERF state. However, the inherent noise of magnetic shield materials is an important factor limiting the measurement sensitivity and accuracy of SERF co-magnetometers. In this paper, both amorphous and nanocrystalline materials were designed and applied as the innermost magnetic shield of an SERF co-magnetometer. Magnetic noise characteristics of different amorphous and nanocrystalline materials used as the internal magnetic shielding layer of the magnetic shielding system were analyzed. In addition, the effects on magnetic noise due to adding aluminum to amorphous and nanocrystalline materials were studied. The experimental results show that compared with an amorphous material, a nanocrystalline material as the inner magnetic shield layer can effectively reduce the magnetic noise and improve the sensitivity and precision of the rotation measurement. Nanocrystalline material is very promising for inner shield composition in SERF co-magnetometers. Furthermore, its ultra-thin structure and low cost have significant application value in the miniaturization of SERF co-magnetometers.

## 1. Introduction

In recent years, a new generation of high-precision atomic sensors, SERF co-magnetometers, have been widely used in Lorentz and CPT violation testing [1,2], magnetic field measurement [3,4], and rotation measurement [5,6]. Their high sensitivity to rotation holds promise as a rotation sensor in next-generation inertial navigation systems [7,8]. High shield factor and low noise in the magnetic shield system is necessary to ensure the normal operation of SERF co-magnetometers [9,10]; however, the inherent noise of the material comprising the magnetic shield system is the main noise term restricting their sensitivity and accuracy [11,12].

Soft magnetic materials are commonly used materials for magnetic shields because of their high magnetic permeability and easy magnetization and demagnetization [13,14,15,16]. For example, μ-metal is often used as a magnetic shielding material because of its high magnetic permeability [17,18,19]. The shield coefficient of magnetic shields made of μ-metal can reach 10${}^{6}$, which can effectively shield against the external environmental magnetic field and quasi-static magnetic field fluctuations [20]. However, a low-resistivity μ-metal shield also produces high intrinsic magnetic noise due to Johnson current noise, which is the main factor limiting the accuracy of SERF co-magnetometers [4,11,21].

In addition to μ-metal, ferrite materials have been increasingly used as magnetic shielding in recent years because they generate less noise than μ-metal [22,23,24,25]. Kornack et al. [26] used manganese–zinc ferrite material in the inner layer of a magnetic shield system, and found that the noise it generated was much lower than that of μ-metal. Subsequently, scholars conducted more in-depth research on the application of manganese–zinc ferrite in SERF co-magnetometers. For example, Fan et al. [27] reported the performance of low-noise ferrites in a K-Rb-${}^{21}$Ne co-magnetometer. Ma et al. [28] studied in detail the magnetic noise model of ferrite and the effect of shielding barrel configuration on noise. However, MnZn ferrite still cannot meet the requirements of low noise, and the manufacture of manganese–zinc ferrite is difficult and expensive. In addition, MnZn ferrite has a low Curie temperature and poor temperature stability. The working temperature of the sensitive element of the co-magnetometer can reach up to 200 degrees, and MnZn ferrite as the innermost magnetic shield produces magnetic field drift. Consequently, it is still necessary to explore new inner shield materials.

Amorphous and nanocrystalline alloys are a new generation of soft magnetic materials with promising application prospects [29]. They possess excellent magnetic properties, such as high saturation magnetic induction, low coercivity, high permeability, and high resistivity [30]. In addition, their unique nanostructure and relatively thin lamination can suppress eddy current loss, and is widely used in high-frequency fields, such as manufacturing transformer cores, power-supply switching and power electronic components [31,32]. Among them, iron-based amorphous and nanocrystalline alloys can be used as materials for making magnetic shields because of their high magnetic permeability [33]. Fu et al. applied nanocrystals to an inner-layer magnetic shield, proving the application potential of nanocrystals in the magnetic shield of SERF co-magnetometers [34]. However, the application and performance of amorphous and nanocrystalline alloys in SERF co-magnetometer magnetic shields has not been deeply studied.

In this paper, inner magnetic shields made of iron-based amorphous and nanocrystalline alloys for high-precision SERF co-magnetometers are proposed and designed. Firstly, a more accurate magnetic noise-error response model for the SERF co-magnetometer was established according to the Bloch equation. Next, the magnetic noise of a magnetic shield comprising amorphous and nanocrystalline alloys was measured separately using a SERF co-magnetometer. Lateral comparison of the magnetic noise generated by amorphous and nanocrystalline inner layer magnetic shields was conducted. At the same time, the magnetic noise of the inner magnetic shield comprising different layers of amorphous material and layers of nanocrystalline material were compared longitudinally. The magnetic noise characteristics of an amorphous inner magnetic shield and a nanocrystalline inner magnetic shield were studied and analyzed. Finally, the magnetic noise characteristics of amorphous and nanocrystalline inner magnetic shields with added aluminum were studied. Our theory and experiments show that nanocrystalline shields have lower thermal magnetization noise and better temperature stability compared to amorphous shields, making them a more favorable choice for the innermost shield material in SERF co-magnetometers. This is of great significance for miniaturization and engineering of low-noise magnetic shield systems for SERF co-magnetometers in future.

## 2. Magnetic Shield Noise Error Model Analysis

The time evolution of the coupled spin ensemble in a SERF co-magnetometer can be solved analytically with a complete set of Bloch equations [35,36]:(1)$$\begin{array}{cc} \; & \frac{\partial P^{e}}{\partial t}=\frac{\gamma^{e}}{Q\left(P^{e}\right)}\left(B+\lambda M^{n}P^{n}+L\right)\times P^{e}-\Omega \times P^{e}+\frac{\left(R_{p}s_{p}+R_{m}s_{m}+R_{se}^{en}P^{n}-R_{tot}^{e}P^{e}\right)}{Q\left(P^{e}\right)}\\ \; & \frac{\partial P^{n}}{\partial t}=\gamma^{n}\left(B+\lambda M^{e}P^{e}\right)\times P^{n}-\Omega \times P^{n}+R_{se}^{ne}P^{e}-R_{tot}^{n}P^{n} \end{array}$$

Here, $P^{e}$ and $P^{n}$ are the electron spin polarization and nuclear spin polarization, respectively. $\gamma^{e}$ and $\gamma^{n}$ are the gyromagnetic ratios of the electron and the nuclear spins, respectively. Q is the electron slowing-down factor, B is the ambient magnetic field vector, L is AC-Stark shift, Ω is the measured rotation vector. $s_{p}$ and $s_{m}$ are the photon spin vectors of the pump and probe beams, oriented along the z and x axes, respectively. $R_{tot}^{e}$ is sum of transverse relaxation of electron spin, $R_{se}^{en}$ is the spin-exchange rate from collisions with the nuclear spins by the electron spins, $R_{se}^{ne}$ is the spin-exchange rate from collisions with the electron spins by the nuclear spins, and $R_{tot}^{n}$ is the total relaxation rate of nuclear spins. $\lambda M^{e}$ and $\lambda M^{n}$ are the electron spin magnetic moment of alkali metal atoms and the nuclear spin magnetic moment of inert gas atoms respectively.

When the co-magnetometer works under steady-state conditions, The input and output of the system can be approximated in the following linear form
(2)$$\dot{X}=AX+WU$$

Here, the state vector **X** can be written as $X={\left[P_{x}^{e},P_{y}^{e},P_{x}^{n},P_{y}^{n}\right]}^{T}$, and the input vector U can be written as $U={\left[\Omega_{x},\Omega_{y},B_{x},B_{y}\right]}^{T}$. The matrix A can be written as:(3)$$A=\left[\begin{array}{cccc} -\frac{R_{\mathrm{tot}\;}^{e}}{Q} & \frac{\gamma^{e}B_{z}^{e}}{Q} & \frac{R_{\mathrm{se}}^{\mathrm{en}}}{Q} & \frac{\gamma^{e}B_{z}^{n}P_{z}^{e}}{QP_{z}^{n}}\\ -\frac{\gamma^{e}B_{z}^{e}}{Q} & -\frac{R_{\mathrm{tot}}}{Q} & -\frac{\gamma^{e}B_{z}^{n}P_{z}^{e}}{QP_{z}^{n}} & \frac{R_{\mathrm{se}}^{\mathrm{en}}}{Q}\\ R_{\mathrm{se}}^{\mathrm{ne}} & \frac{\gamma^{n}P_{z}^{n}B_{z}^{e}}{P_{z}^{e}} & -R_{\mathrm{tot}}^{n} & \gamma^{n}B_{z}^{n}\\ -\frac{\gamma^{n}B_{z}^{e}P_{z}^{n}}{P_{z}^{e}} & R_{\mathrm{se}}^{\mathrm{ne}} & -\gamma^{n}B_{z}^{n} & -R_{\mathrm{tot}}^{n} \end{array}\right]$$

The matrix W can be written as
(4)$$W=\left[\begin{array}{cccc} 0 & -P_{z}^{e} & 0 & \frac{P_{z}^{e}\gamma^{e}}{Q}\\ P_{z}^{e} & 0 & -\frac{P_{z}^{e}\gamma^{e}}{Q} & 0\\ 0 & -P_{=}^{n} & 0 & P_{z}^{n}\gamma^{n}\\ P_{z}^{n} & 0 & -P_{z}^{n}\gamma^{n} & 0 \end{array}\right]$$

Therefore, the transfer function that characterizes the input–output relationship in the system at steady state can be expressed as $G\left(s\right)={\left(sI-A\right)}^{-1}W$. After calculation and simplification, the transfer function between input $B_{x}$ and output $P_{x}^{e}$ can be expressed as [12]
(5)$$\begin{array}{cc} G_{Bx}\left(s\right) & =P_{x}^{e}\left(s\right)/B_{x}\left(s\right)=\frac{k_{Bx}\left(s-\omega_{Bx1}\right)\left(s-\omega_{Bx2}\right)}{\left[{\left(s-\varphi_{1}\right)}^{2}+\omega_{1}^{2}\right]\left[{\left(s-\varphi_{2}\right)}^{2}+\omega_{2}^{2}\right]} \end{array}$$

Here, the zeros are
(6)$$\omega_{Bx1}{=}_{-}\frac{P_{z}^{e}R_{tot}^{n}\gamma^{e}+P_{z}^{n}R_{se}^{en}\gamma^{n}}{P_{z}^{e}\gamma^{e}}$$
(7)$$\omega_{Bx2}=\frac{B_{z}^{e}P_{z}^{e}R_{tot}^{n}\gamma^{e}+B_{z}^{e}P_{z}^{n}R_{se}^{en}\gamma^{n}+B_{z}^{n}P_{z}^{e}R_{tot}^{e}\gamma^{n}}{P_{z}^{e}\left(B_{z}^{e}\gamma^{e}+B_{z}^{n}\gamma^{n}Q\right)}$$

There are two separate oscillations with different frequencies $\varphi_{1}$, $\varphi_{2}$, and decay rates $\omega_{1}$, $\omega_{2}$. $B_{z}^{n}$ is the effective field for the z-direction nuclear magnetization and $B_{z}^{e}$ is the effective field for the z-direction electron magnetization. The co-magnetometer is sensitive to $B_{x}$ when they are sensitive to $\Omega_{y}$, which is why the performance of gyroscopes based on SERF co-magnetometers is limited by magnetic noise. Therefore, low-noise shielding is very important for SERF co-magnetometers.

When the SERF co-magnetometer is sensitively rotated, the fluctuation of $B_{x}$ will cause the SERF co-magnetometer to produce bias drift, and the navigation error will accumulate over time. Therefore, it is necessary to establish a magnetic noise error model to analyze the equivalent error generated by magnetic noise. To obtain the power spectrum of the rotating signal of the SERF co-magnetometer, it is necessary to obtain the power spectral density of the static test data and the scale factor of the rotation, and then divide the static power spectral density by the scale factor to obtain the final signal power spectrum. At this point, the stochastic process associated with the system output signal can be regarded as a stationary stochastic process. According to the stochastic process analysis theory, the bilateral power spectral density (PSD) of the output signal when a stationary stochastic process is superimposed on a linear system can be expressed as:(8)$$S_{Y}\left(\omega \right)=S_{X}\left(\omega \right){\left|H\left(\omega \right)\right|}^{2}$$
where $S_{X}\left(\omega \right)$ is the bilateral PSD of the input process, $S_{Y}\left(\omega \right)$ is the bilateral PSD of the output process, ${\left|H\left(\omega \right)\right|}^{2}$ is the power transfer function of the linear system, and |H(ω)| is the amplitude–frequency response of the system. The steady state rotation rate in the time domain can be expressed as:(9)$$\Omega_{y}\left(t\right)=K_{\Omega_{y}}P_{x}^{e}\left(t\right)$$

The $\Omega_{y}$ can be described in the Laplace transform domain as:(10)$$\begin{array}{cc} \Omega_{y}\left(s\right) & =K_{\Omega_{y}}G_{Bx}\left(s\right)B_{x}\left(s\right)=\frac{K_{\Omega_{y}}K_{Bx}\left(s-z_{Bx1}\right)\left(s-z_{Bx2}\right)B_{x}\left(s\right)}{\left[{\left(s-\Gamma_{1}\right)}^{2}+\omega_{1}^{2}\right]\left[{\left(s-\Gamma_{2}\right)}^{2}+\omega_{2}^{2}\right]}. \end{array}$$
where the scale factor $K_{\Omega_{y}}$ can be expressed as:(11)$$K_{\Omega_{y}}=\frac{{\left(\gamma^{e}B^{e}R_{\mathrm{tot}\;}^{n}+\gamma^{n}B^{n}R_{\mathrm{tot}\;}^{e}\right)}^{2}+{\left(R_{\mathrm{tot}\;}^{n}R_{\mathrm{tot}\;}^{e}\right)}^{2}}{P_{z}^{e}\gamma^{e}R_{\mathrm{tot}\;}^{e}{\left(B^{n}\right)}^{2}\gamma^{n}}$$

In the SERF co-magnetometer inertial measurement system, the hysteresis loss magnetic noise and eddy current loss magnetic noise of the innermost shielding cylinder are uncorrelated, therefore the total angular velocity PSD caused by the two magnetic noises can be expressed as:(12)$$S_{\Omega y}^{T}\left(\omega \right)=S_{\Omega y}^{\mathrm{magn}\;}\left(\omega \right)+S_{\Omega y}^{\mathrm{eddy}\;}\left(\omega \right)$$

Among them, $S_{\Omega y}^{\mathrm{magn}\;}\left(\omega \right)$ is the angular velocity PSD corresponding to the hysteresis loss magnetic noise, and $S_{\Omega y}^{\mathrm{eddy}\;}\left(\omega \right)$ is the angular velocity PSD corresponding to the eddy current loss magnetic noise. When the co-magnetometer is in a steady state, the steady-state PSD can be expressed as:(13)$$\begin{array}{cc} \; & S_{\Omega }^{\mathrm{magn}\;}\left(\omega \right)=S^{\mathrm{magn}\;}\left(\omega \right){\left|G_{Bx}\left(0\right)\right|}^{2}K_{\Omega_{y}}^{2}\\ \; & S_{\Omega }^{\mathrm{eddy}\;}\left(\omega \right)=S^{\mathrm{eddy}\;}\left(\omega \right){\left|G_{Bx}\left(0\right)\right|}^{2}K_{\Omega_{y}}^{2} \end{array}$$
where $S^{\mathrm{magn}\;}\left(\omega \right)=C_{\mathrm{magn}\;}^{2}/2\omega$ and $S^{\mathrm{eddy}\;}\left(\omega \right)=C_{\mathrm{eddy}\;}^{2}/2$ are the bilateral PSD of hysteresis loss noise and eddy current loss noise, respectively. $C_{\mathrm{magn}\;}$ and $C_{\mathrm{eddy}\;}$ are the noise figures of hysteresis loss noise and eddy current loss noise, respectively.

The relationship between the Allan variance and the bilateral PSD of the angular velocity is:(14)$$\sigma^{2}\left(T\right)=\frac{4}{\pi T}\int_{0}^{\infty }S_{\Omega }\left(\frac{\mu }{\pi T}\right)\frac{\sin^{4}u}{u^{2}}\;du$$

Here u=πνT, where ν is frequency and *T* is the group time.

Bringing $S_{\Omega y}^{\mathrm{magn}\;}\left(\omega \right)$ and $S_{\Omega y}^{\mathrm{eddy}\;}\left(\omega \right)$ into Equation (Equation 14) produces their Allan variance expression:(15)$${\left[\sigma_{\Omega }^{\mathrm{magn}}\left(T\right)\right]}^{2}=\frac{C_{\mathrm{magn}\;}^{2}{\left|G_{Bx}\left(0\right)\right|}^{2}}{\pi }\ln2\left(T\gg 1/\nu_{0}\right)$$
(16)$${\left[\sigma_{\Omega }^{\mathrm{eddy}\;}\left(T\right)\right]}^{2}=\frac{C_{\mathrm{eddy}\;}^{2}{\left|G_{Bx}\left(0\right)\right|}^{2}}{2T}$$

Here, $\nu_{0}$ represents the corner frequency of the hysteresis loss magnetic noise. The analysis shows that the Allan variance of the low-frequency angular velocity corresponding to the hysteresis loss magnetic noise does not change with the correlation time. Eddy current loss noise is inversely proportional to T and has a −1/2 slope with respect to T in a log–log plot.

Therefore, according to the above analysis, it is not difficult to see that we can use the magnetic noise PSD, the scale coefficient and the amplitude–frequency response to Bx to evaluate the performance of the inner magnetic shield.

## 3. Experimental Setup

The K-Rb-${}^{21}\mathrm{Ne}$ co-magnetometer was used to conduct the experiment, and the schematic diagram of the structure is shown in Figure 1. The constructed K-Rb-${}^{21}\mathrm{Ne}$ co-magnetometer structural components are similar to previous work [12]. The magnetic shield system of the K-Rb-${}^{21}\mathrm{Ne}$ co-magnetometer consists of three shield layers, of which the outer two layers are μ-metal. μ-metal has a high shielding coefficient and is used to shield the interference of geomagnetism and external magnetic fields. The inner shielding layer is to shield the magnetic field and reduce the magnetic noise in the shielding system. In order to ensure that the device works in a low magnetic field environment, the magnetic shield system must be degaussed before the device is started. After degaussing, the residual magnetic field in the shielded barrel is compensated with a high-precision three-axis magnetic field coil. The sensitive core is a 10 mm diameter cell made of GE180 aluminosilicate glass. The cell is placed in a boron nitride oven, which is heated using a heating coil connected to the oven, allowing it to operate at a maximum temperature exceeding 200 °C.

The experimental setup shown in Figure 1 is a structural diagram of an experimental setup we built, which can be used to sense magnetic fields or to sense rotation. The difference between the two devices is that different atoms are used in the cell. When the device operates as a SERF magnetometer sensitive to magnetic field, the cell contains the natural-abundance K and Rb alkali metal mixture, 1.9 atm of ${}^{4}He$ buffer gas, and 50 Torr $N_{2}$ quench gas. When the device operates as a SERF co-magnetometer sensitive to rotation, the cell contains a natural-abundance K and Rb alkali metal mixture, 2 atm of ${}^{21}\mathrm{Ne}$ and 50 Torr $N_{2}$ quench gas.The circular-polarized laser light propagating along the z-axis is emitted by a distributed Bragg reflector (DBR) diode laser, interacting with the D1 resonance line of K atoms to realize the polarization of K atoms. The probe laser is a non-resonant linear-polarized light emitted along the x-axis to detect the atomic spin precession of the Rb atoms along the x-axis. A signal generator is used to control the triaxial magnetic field coil in order to apply excitation to the magnetic field. The ratio of alkali metal density inside the cell is K:Rb = 1:90 at 180 °C.

Figure 2 shows the inner magnetic shields of the magnetometer fabricated for this experiment. Amorphous and nanocrystalline shields are made of Fe-based amorphous and nanocrystalline films attached to a support frame made of non-magnetic PTFE. The thickness of the support frame made of PTFE is 5 mm, the outer radius of the barrel is 47 mm, and the barrel height is 120 mm. The test results of the inner shield layer of amorphous alloy and the inner shield layer of nanocrystalline alloy in SERF magnetometer was compared to analyze the influence of the inner shield magnetic noise on the measurement accuracy of SERF co-magnetometer.

2-layer amorphous inner magnetic shield4-layer amorphous inner magnetic shield2-layer nanocrystalline inner magnetic shield4-layer nanocrystalline inner magnetic shield2-layer amorphous+aluminum inner magnetic shield4-layer amorphous+aluminum inner magnetic shield2-layer nanocrystalline+aluminum inner magnetic shield4-layer nanocrystalline+aluminum inner magnetic shield

We designate these eight different inner magnetic shielding cylinder schemes as schemes 1–8.

As shown in Figure 3, the demagnetization coil structure is composed of six annular coils wound around the magnetic shield system. The coil can generate magnetic flux in a closed-loop mode in the shield layer, which can demagnetize the magnetic shield system. Further, Figure 3b is the size of the three-layer shielding cylinder. As the outer shield layer of the magnetic shield system, the main function of the two-layer permalloy is to shield the external magnetic field. The outermost layer of permalloy has a height of 180 mm and an outer diameter of 128 mm, and the second outer layer of permalloy has a height of 150 mm and an outer diameter of 108 mm. The innermost layer is an inner shielding layer composed of amorphous/nanocrystalline material and PTFE. PTFE cylinder with a thickness of 6 mm, amorphous/nanocrystalline material wound on the outside of the PTFE cylinder. The reason for using PTFE is that amorphous/nanocrystalline material cannot be fixed as ultrasoft magnetic materials, requiring a supporting skeleton. In addition, PTFE is a non-magnetic material and will not generate magnetic field interference.

## 4. Results and Discussion

According to the analysis results in Section 2, the magnetic noise of the magnetic shield system mainly depends on the innermost magnetic shield layer. The low-frequency magnetic noise we focus on mainly includes the magnetic noise caused by hysteresis loss and eddy current loss. According to Equations (Equation 15) and (Equation 16), it is known that the noise due to hysteresis loss is similar to 1/f noise, and the noise due to eddy current loss is similar to white noise.

The magnetic noise characteristics of the inner shield layers of eight different schemes were measured in situ with a K-Rb-${}^{21}\mathrm{Ne}$ SERF magnetometer. Table 1 shows the test results of the residual magnetic fields in the three directions of the center of the shielding system under the inner shield layer of Scheme 1 to Scheme 8 after the winding is demagnetized. The main purpose of the design of the inner shield is to suppress the magnetic noise felt by the central air chamber of the magnetic shielding system. On the one hand, it is necessary to suppress the magnetic noise caused by the outer magnetic shield. The second is to suppress the magnetic noise generated by the inner shielding material itself. This is also the reason why SERF co-magnetometers need to choose low noise and high permeability inner shields.

The degaussing current is generated by a laptop, a compact DAQ module, and a power amplifier. The degaussing current is about 60A. It can be seen that with the same degaussing method, the residual magnetic flux density of nanocrystals is significantly smaller than that of the amorphous material. The small residual flux density can make the angle measurement more sensitive and stable.

Figure 4 shows the measurement results of magnetic noise of the SERF magnetometer under the inner magnetic shield layer of amorphous alloys and nanocrystalline alloys at frequencies below 100 Hz. It can be seen that the noise of the two amorphous inner shield layers is lower than the four amorphous inner shield layers. Similarly, the noise of the two nanocrystalline inner shield layers is lower than the four nanocrystalline inner shield layers. Therefore, the amorphous and nanocrystalline materials have similar properties when used as the inner magnetic shield. With the increasing number of layers, the magnetic noise of the amorphous and nanocrystalline inner magnetic shielding becomes greater. It can be seen from Figure 4 that the interference of the small peaks that appear is around 50 Hz, and is caused by power frequency fluctuations. Since the SERF co-magnetometer is used for rotation measurement, the further the magnetic noise is below 1 Hz, the higher the measurement accuracy of rotation. It can be seen that the magnetic noise level of nanocrystalline alloy is much better than that of amorphous alloy at frequencies lower than 1 Hz. Therefore, for rotation measurement, nanocrystalline alloys as the inner magnetic shield layer are better than amorphous.

In addition, the magnetic noise characteristics of amorphous and nanocrystalline magnetic shield inner layers with aluminum addition were studied. Figure 5 shows magnetic noise measurement results of the SERF magnetometer under inner magnetic shield layers of different (Figure 5a) amorphous+aluminum alloys and (Figure 5b) nanocrystalline +aluminum alloys. It can be seen from Figure 5 that with the increase of the number of layers, the magnetic noise of both amorphous and nanocrystalline material decreases, and the magnetic noise characteristics change. The magnetic noise of amorphous material after adding aluminum is lower than that without aluminum, but it is still higher than that of nanocrystalline material. However, after adding aluminum to nanocrystalline material, compared with no aluminum, the magnetic noise below 1Hz has increased; thus, adding aluminum to nanocrystalline material is not a good idea.

Table 2 shows the magnetic noise values of the eight schemes at 1 Hz, from which it can be seen that the 2-layer nanocrystalline alloy is the scheme with the lowest noise, and the magnetic noise is 14.17 fT/Hz${}^{1/2}$. Among the four kinds of amorphous inner magnetic shields, the best magnetic noise at 1 Hz is scheme 8, having a magnetic noise value of 54.13 fT/Hz${}^{1/2}$.

According to the data measured thus far, nanocrystalline material with fewer layers is a better choice for inner magnetic shielding. However, the research on the magnetic noise characteristics of nanocrystalline is not adequate at the present time. Later, detailed magnetic noise modeling will be carried out to analyze its characteristics.

## 5. Conclusions

This paper analyzes the magnetic noise characteristics of amorphous and nanocrystalline inner shields used in SERF co-magnetometer combined magnetic shielding systems. Comparing the magnetic noise of amorphous and nanocrystalline materials with different layers, it is found that the magnetic noise increases with the increase of layers. Further, the magnetic noise of nanocrystalline material is markedly lower than that of amorphous material when the frequency is lower than 1 Hz. In addition, aluminum was added to amorphous and nanocrystalline materials. It was found that the magnetic noise decreases with the increased number of layers after adding aluminum, which requires further research into its characteristics. The research and findings in this paper are not only instructive for the exploration of miniaturized low-noise magnetic shield systems more suitable for high-precision SERF co-magnetometers, it also possesses important guiding significance for other high-precision atomic sensors that require magnetic shield systems, such as atomic clocks and nuclear magnetic resonance gyroscopes.

## Figures and Tables

**Figure 1 materials-15-08267-f001:**
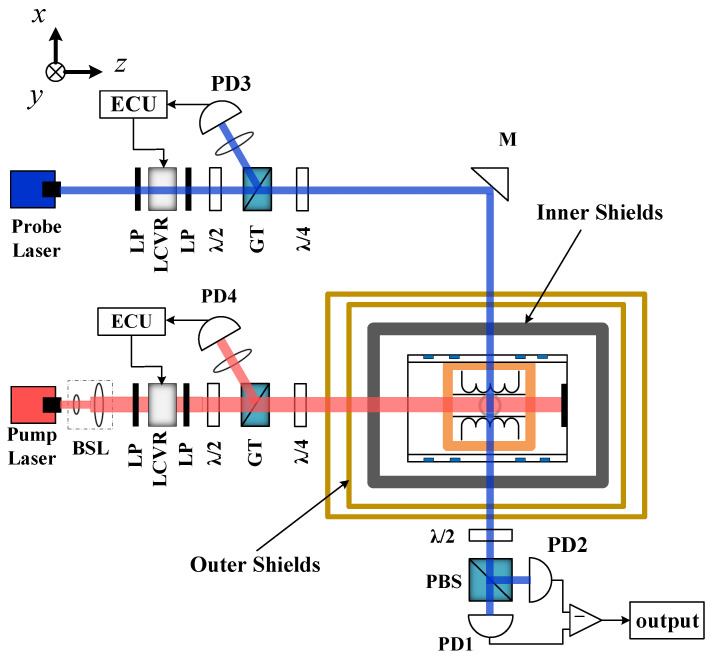
Schematic of K-Rb-${}^{21}\mathrm{Ne}$ co-magnetometer. BSL: beam shaping lenses. LP: linear polarizer. LCVR: liquid crystal variable retarder. λ/2: half wave plate. GT: Glan–Taylor polarizer. λ/4: quarter wave plate. PBS: polarizing beam splitter. M: reflection mirror. PD: photodiode. ECU: electronic control unit. The magnetic shield system of common magnetic force consists of three shield layers. The outer shield consists of two 2 mm thick μ-metal shields with outer radii of 54 mm and 64 mm and lengths of 150 mm and 180 mm, respectively. The inner shield is the focus of our research.

**Figure 2 materials-15-08267-f002:**
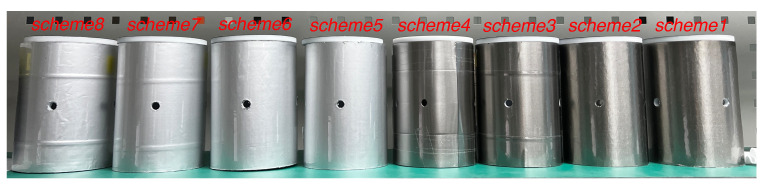
Amorphous and nanocrystalline inner magnetic shield layers.

**Figure 3 materials-15-08267-f003:**
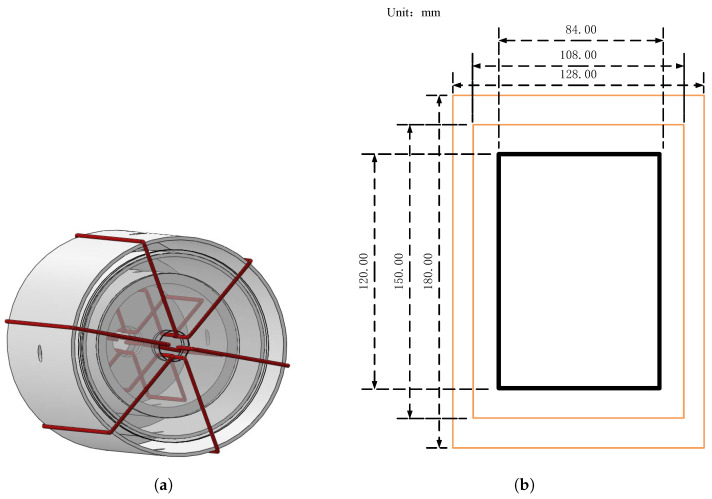
Internal structure of magnetic shield system composed of three shield layers of SERF co-magnetometer and method of wound demagnetizing coil. (**a**) 3D structure diagram (**b**) Plane dimension drawing.

**Figure 4 materials-15-08267-f004:**
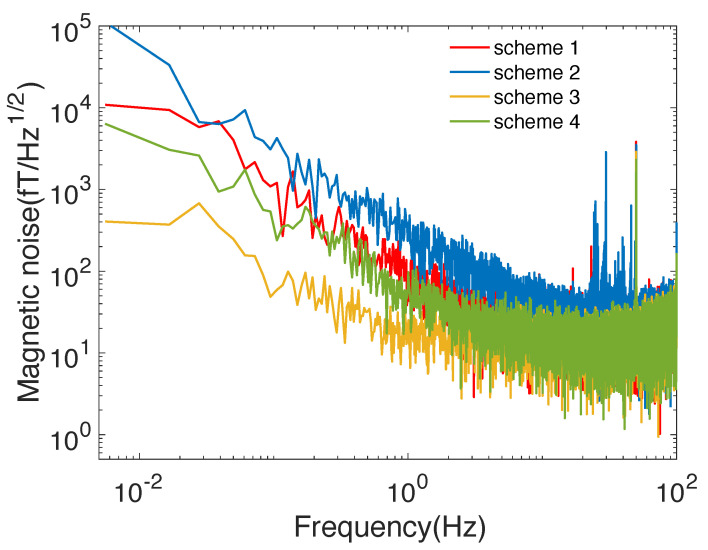
Magnetic noise measurement results of the SERF magnetometer under inner magnetic shield layers of different amorphous alloys and nanocrystalline alloys.

**Figure 5 materials-15-08267-f005:**
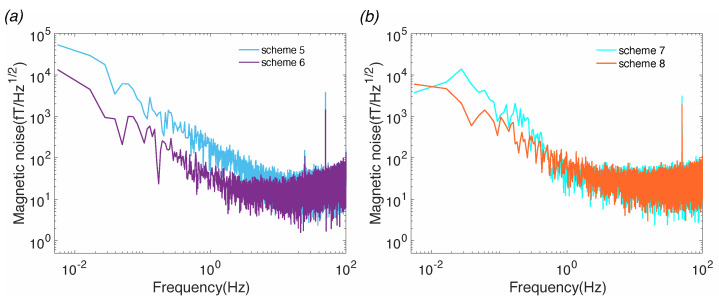
Magnetic noise measurement results of the SERF magnetometer under inner magnetic shield layers of different (**a**) amorphous + aluminum alloys, and (**b**) nanocrystalline + aluminum alloys.

**Table 1 materials-15-08267-t001:** Triaxial residual magnetic field after demagnetization of combined magnetic shielding system with different amorphous and nanocrystalline material as inner shield.

	$B_{x}$ (nT)	$B_{y}$ (nT)	$B_{z}$ (nT)
without shielding	9385	29,665	29,850
Scheme 1	3	3.78	11.25
Scheme 2	1.3	2.4	12.45
Scheme 3	0.56	2.4	0.3
Scheme 4	0.86	1.08	0.375
Scheme 5	7.2	6.34	11.1
Scheme 6	13.14	5.54	0.9
Scheme 7	0.5	1.92	0.33
Scheme 8	0.46	0.46	0.855

**Table 2 materials-15-08267-t002:** Magnetic noise value of schemes 1–8 at 1 Hz.

	1 Hz fT/Hz${}^{1/2}$
Scheme 1	68.93
Scheme 2	239.65
Scheme 3	147.98
Scheme 4	28.62
Scheme 5	14.17
Scheme 6	46.59
Scheme 7	54.81
Scheme 8	54.13

## Data Availability

Not applicable.

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
