# Peer review of "Study on the Magnetic Noise Characteristics of Amorphous and Nanocrystalline Inner Magnetic Shield Layers of SERF Co-Magnetometer"

_materials, 2022, doi:10.3390/ma15228267_

Round 1
Reviewer 1 Report
The paper presents a study on the magnetic noise characteristics of amorphous and nanocrystalline inner magnetic shield layers of SERF Co-magnetometer. According to the reviewer’s opinion, the paper is well-structured and clear. The topic is interesting and falls within the aim of the journal. In addition, the results are well-presented and could be helpful to further develop the same topic. Therefore, the paper can be accepted for publication in the current form.
Author Response
Thank you very much for your affirmation of my article.
Reviewer 2 Report
Paper can be accepted after the following corrections:
1. Please specify the detailed composition of the amorphous and nanocrystalline alloy. Details concerning the thermal treatment of the materials should also be specified.
2. Please provide the information concerning the value of magnetic field without shielding (Table 1)
3. Uncertainties of both magnetic field measurements and noise measurements should be at least roughly specified.
4. Please note that the degaussing process is probably negligible for ultrasoft magnetic materials (amorphous and nanocrystaline alloys).
5. Conclusions should be developed and presented in the more quantitative way.
Reviewer 3 Report
The paper "Study on the Magnetic Noise Characteristics of Amorphous and Nanocrystalline Inner Magnetic Shield Layers of SERF Co-Magnetometer" by Ye Liu et al presents the results of analysis of the the magnetic noise characteristics of amorphous and nanocrystalline inner shields used in high-accuracy SERF co-magnetometer. It is found that the magnetic noise increases with the increase of layers by comparing the magnetic noise of amorphous and nanocrystalline with different layers. In addition, it is found that the magnetic noise decreases with the increase of the number of layers after adding aluminum. The presented results could be used for high-precision SERF co-magnetometer, but also significant for other atomic sensors that need magnetic shield system. The manuscript can be published in journal after answering some minor questions, which are listed below.
1) On Fig.4 noise measurements shows many peaks at the higher frequency. What caused this? The description should be added.
2) How the temperature is affected on distribution of magnetic noise in that kind of systems? The description should be added.
3) If the Auhthors in the introduction use the idea of magnetometer principles, the spin-wave spectroscopy in ferromagnetic films should be provided. Thus the Ref[JETP LETTERS 108 (5), pp.312-317 (2018)] and [JETP LETTERS 102 (3), pp.142-147(2015)] should be added along with the Refs[1-19]
4) In Figure 3, it is recommended to indicate the characteristic dimensions and distances between the layers, for a better understanding of the geometry of the structure.
Round 2
Reviewer 2 Report
The paper was corrected and can be accepted in the present state.
Reviewer 3 Report
The paper presents a study on the magnetic noise characteristics of amorphous and nanocrystalline inner magnetic shield layers of SERF Co-magnetometer.
According to the Authors' response and new version of the manuscrip the paper can be accepted for publication.